# Canopy Transpiration and Stomatal Conductance Dynamics of *Ulmus pumila* L. and *Caragana korshinskii* Kom. Plantations on the Bashang Plateau, China



Yu Zhang [1,2,3], Wei Li [4,5,6,*], Haiming Yan [4,5,6], Baoni Xie [4,5,6], Jianxia Zhao [7], Nan Wang [4] and Xiaomeng Wang [4]

1 School of Geographical Sciences, Hebei Normal University, Shijiazhuang 050024, China; yuzhang89@mail.bnu.edu.cn
2 Hebei Key Laboratory of Environmental Change and Ecological Construction, Shijiazhuang 050024, China
3 Hebei Technology Innovation Center for Remote Sensing Identification of Environmental Change, Shijiazhuang 050024, China
4 School of Land Science and Space Planning, Hebei GEO University, Shijiazhuang 050031, China; haiming.yan@hgu.edu.cn (H.Y.); xbn-feya@nwafu.edu.cn (B.X.); wn305099@163.com (N.W.); wxm975000@163.com (X.W.)
5 International Science and Technology Cooperation Base of Hebei Province, Hebei International Joint Research Center for Remote Sensing of Agricultural Drought Monitoring, Hebei GEO University, Shijiazhuang 050031, China
6 Hebei Province Collaborative Innovation Center for Sustainable Utilization of Water Resources and Optimization of Industrial Structure, Hebei GEO University, Shijiazhuang 050031, China
7 College of Geography and Land Engineering, Yuxi Normal University, Yuxi 653100, China; zjx@yxnu.edu.cn
* Correspondence: weil87land@hgu.edu.cn

**Abstract:** Constructing protective forests to control water and soil erosion is an effective measure to address land degradation in the Bashang Plateau of North China, but forest dieback has occurred frequently due to severe water deficits in recent decades. However, transpiration dynamics and their biophysical control factors under various soil water contents for different forest functional types are still unknown. Here, canopy transpiration and stomatal conductance of a 38-year-old *Ulmus pumila* L. and a 20-year-old *Caragana korshinskii* Kom. were quantified using the sap flow method, while simultaneously monitoring the meteorological and soil water content. The results showed that canopy transpiration averaged $0.55 \pm 0.34$ mm d$^{-1}$ and $0.66 \pm 0.32$ mm d$^{-1}$ for *U. pumila*, and was $0.74 \pm 0.26$ mm d$^{-1}$ and $0.77 \pm 0.24$ mm d$^{-1}$ for *C. korshinskii* in 2020 and 2021, respectively. The sensitivity of canopy transpiration to vapor pressure deficit (*VPD*) decreased as soil water stress increased for both species, indicating that the transpiration process is significantly affected by soil drought. Additionally, canopy stomatal conductance averaged $1.03 \pm 0.91$ mm s$^{-1}$ and $1.34 \pm 1.22$ mm s$^{-1}$ for *U. pumila*, and was $1.46 \pm 0.90$ mm s$^{-1}$ and $1.51 \pm 1.06$ mm s$^{-1}$ for *C. korshinskii* in 2020 and 2021, respectively. The low values of the decoupling coefficient ($\Omega$) showed that canopy and atmosphere were well coupled for both species. Stomatal sensitivity to *VPD* decreased with decreasing soil water content, indicating that both *U. pumila* and *C. korshinskii* maintained a water-saving strategy under the stressed water conditions. Our results enable better understanding of transpiration dynamics and water-use strategies of different forest functional types in the Bashang Plateau, which will provide important insights for planted forests management and ecosystem stability under future climate changes.

**Keywords:** Bashang Plateau; transpiration; canopy stomatal conductance; soil water; afforestation

## 1. Introduction

Afforestation is an important ecological measure for restoring fragile and degraded land. China has the largest afforested area in the world, with the area of planted forests accounting for about 23% of the global plantation area [1]. The area of planted forests

in China now stands at about $69.33 \times 10^6$ ha. Planted forests can effectively contribute to improving the functions of water and soil conservation, sand-fixing, carbon storage, and adjusting the microclimate of weak ecological environment systems [2,3]. However, under the influence of global climate change-induced drought and human activities, these planted forests, especially in arid and semiarid regions, have suffered severe canopy dieback and mortality approximately 30–35 years after planting [4,5]. Immense afforestation may consume too much water and cause a severe water deficit, and, compared with native plantation species, introduced plantation species cause a greater reduction in soil water availability, negatively impacting ecosystem services and functions [6,7]. Transpiration and related water-use strategies are fundamental to understanding the physiological processes of plantations and play a vital role in their survival and growth, especially in semiarid and arid regions, where water availability is greatly affected by the increased frequency and intensity of droughts [8,9]. However, previous studies have shown that the effects of excessive water loss caused by transpiration and its environmental driving factors differ significantly, resulting in different growth performances among different functional types [10–12]. Therefore, it is imperative to understand the transpiration processes and dynamics of planted tree and shrub species and how these processes change from their normal functions.

Transpiration, the plant process in which water is consumed from the soil and released into the atmosphere, is an important parameter in understanding forest hydrological processes and further determining the forest water balance [13,14]. The accurate quantification of forest transpiration is essential for tree physiology and ecohydrology [15,16]. Based on their advantages of simple operation, relatively low cost, fewer limitations in environmental conditions, and increasingly robust features, sap flow measurements have been considered the most practical method to estimate tree transpiration at both the individual and stand scales [17,18]. Environmental factors, including solar radiation, air temperature, wind speed, vapor pressure deficit (*VPD*), and soil water content, have significant impacts on the transpiration process. These abiotic factors can be divided into two aspects: evaporative demand (potential evapotranspiration, PET) and water supply [19]. In semiarid and arid regions, soil water availability is a critical factor for plantation growth and vegetation productivity due to its influence on plant transpiration [20]. Previous studies showed that soil water stress in plantations occurs when relative extractable water content (REW) is below a threshold of 0.4 [21,22]. A linear or nonlinear relationship, including exponential and polynomial functions, was found between transpiration and evaporative demand when REW > 0.4, whereas, when REW < 0.4, increased soil water stress influenced transpiration, which was expressed following exponential and polynomial functions for evaporative demand for different planted forests [23]. Moreover, transpiration is regulated via canopy stomatal conductance ($G_c$) in response to variations in *VPD* and soil water content. High *VPD* causes a decrease in canopy stomatal conductance. In order to prevent irreversible damage to tree hydraulic traits, transpiration then gradually increases nonlinearly with increasing *VPD*, and is maintained or decreases when the canopy stomata begin to close. However, different specific nonlinear functions, such as exponential functions [24], logarithmic functions, and polynomial functions [25], have been reported for different plantation types in different artificial ecosystems. Additionally, reduced soil water availability may decrease hydraulic conductance from soil to leaves, causing canopy stomatal closure to avoid embolism and hydraulic failure, and thereby reducing tree transpiration. Denham et al. quantified the collaborative influence of soil water content and *VPD* along a hydro-climatological gradient, and indicated that a more sensitive relationship between canopy stomatal conductance and soil water content was found in dry than in wet sites [26]. Most climate models predict that soil droughts will be more frequent and more severe in semiarid and arid regions [27]. However, few studies have explored the impact of environmental factors on transpiration and $G_c$ under different soil water availability conditions, and influenced by climate change-induced drought. Moreover, it is also unclear whether these relationships exhibit interspecific differences among different forest functional types.

The Bashang Plateau is one of the most vulnerable and sensitive portions of northern China, and is a major source of dust and sand storms that affect Beijing and its surrounding areas. Land degradation and desertification are the most serious environmental problems in this area. A series of afforestation projects have been implemented by China's government during the past four decades, e.g., the Grain-for-Green Program, the Three-North Shelterbelt Program, and the Beijing-Tianjin Sand Source Restoration Project, to address land degradation and improve the regional ecological environment [28,29]. Many fast-growing and stress-tolerant species, such as *Populus simonii* Carr, *Pinus sylvestris* var. *mongolica Litv*, *Ulmus pumila* L, and *Caragana korshinskii* Kom were planted to improve vegetation cover and control wind and sand erosion. The degree of desertification on the Bashang Plateau has improved significantly during the past four decades. However, these plantations have frequently suffered canopy dieback and mortality in recent years. The main reason for this is related to water deficiency, which affected plantations' transpiration dynamics and water-use strategies. *U. pumila* is a major native tree species and *C. korshinskii* is an introduced shrub species. Nevertheless, the transpiration processes of the two functional types are still not fully understood, and the biophysical control factors of transpiration under various soil water availability conditions are still unknown. The objectives of this study were to explore: (1) canopy transpiration and stomatal conductance processes during the growing season for *U. pumila* (native species) and *C. korshinskii* (introduced species); and (2) the biophysical mechanisms of canopy transpiration control under different soil water content conditions. The results can provide a deep understanding of the water-use dynamics of planted tree and shrub species, and thus provide scientific guidance for afforestation species selection and optimal allocation.

## 2. Materials and Methods

### 2.1. Site Description

Our study was conducted in the Kangbao pasture region, Kangbao County, Zhangjiakou City, Hebei Province, China (Figure 1). This area is located in the temperate climate zone. The average annual air temperature is about 2.3 °C, and the average annual precipitation is 330 mm, with over 85% falling between May and September. The mean annual potential evapotranspiration is 880 mm and the mean wind speed is 3.15 m s$^{-1}$. The major soil type is chestnut soil. The main growing season runs from May to September. The dominant tree species here are *U. pumila*, *P. simonii*, and *Pinus sylvestris* var. mongolica Litv, and the dominant shrub species are *C. korshinskii*, *Hippophae rhamnoides* Linn, and *Armeniaca sibirica* (L.) Lam.

This experiment was conducted in stands of *U. pumila* (114°47′ E, 42°06′ N, altitude 1325 m) and *C. korshinskii* (114°48′ E, 42°07′ N, altitude 1305 m) in 2020 and 2021. The two sites were approximately 3.0 km apart. A plot having an area of 20 by 20 m was set up in each of the *U. pumila* and *C. korshinskii* stands. The *U. pumila* were about 38 years old and the *C. korshinskii* were about 20 years old. The corresponding maximum leaf area indexes (LAI) were 0.31 and 0.43 m$^2$ m$^{-2}$, respectively. The mean diameter at breast height (DBH, 1.3 m) of *U. pumila* was 16.60 cm and the mean stem basal diameter (SBD, 10–15 cm above the ground) of *C. korshinskii* was 2.38 cm. The physical characteristics of the soil differed between the *U. pumila* and *C. korshinskii* stands. The soil particle size distribution was 39.64% sand (0.05–2 mm), 50.85% silt (0.002–0.05 mm), and 9.69% clay (<0.002 mm) in the *U. pumila* stand, and 52.89% sand, 39.25% silt, and 7.86% clay in the *C. korshinskii* stand. The soil organic matter (SOM) in the *U. pumila* stand at the 0–100 cm depth was 1.34%, and that of the *C. korshinskii* stand was 0.85% (Table 1).

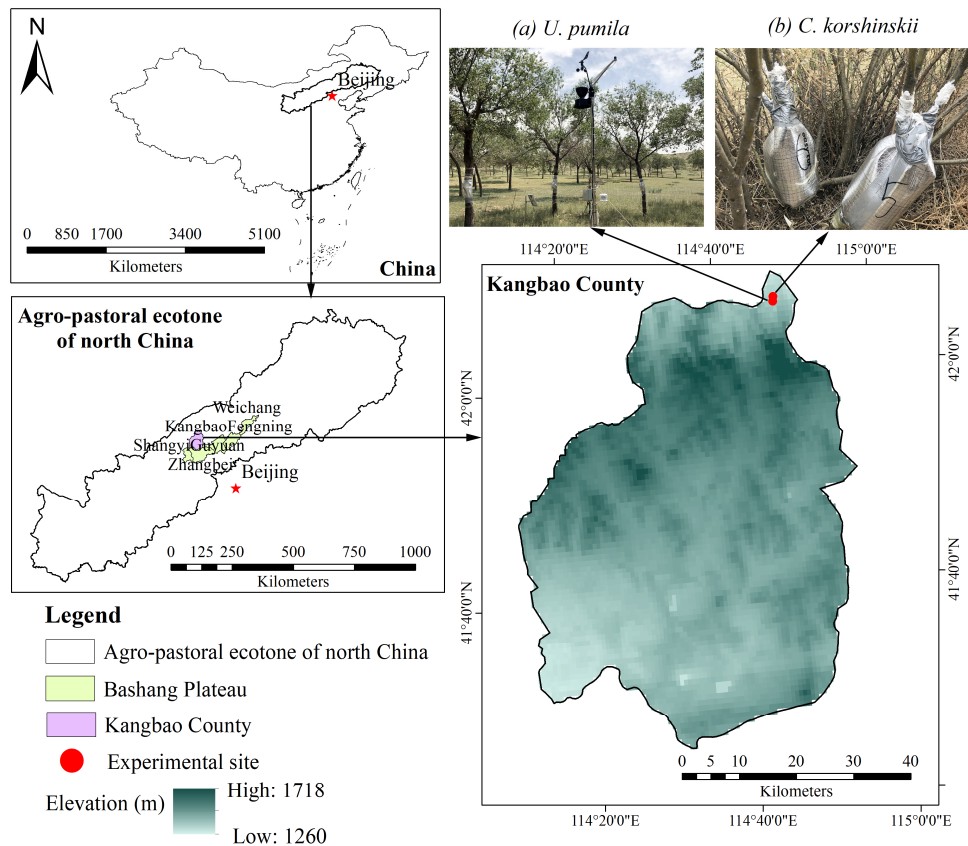

**Figure 1.** Geographical research sites in this study, namely, the (**a**) *U. pumila* site and (**b**) *C. korshinskii* site on the Bashang Plateau.

**Table 1.** Detailed information about vegetation characteristics and soil properties of *U. pumila* and *C. korshinskii* stands.

| Species | Vegetation Characteristics | | | | Soil Properties (%) | | | |
| | Age | DBH/SBD (cm) | SA (cm²) | LAI (m² m⁻²) | Soil Particle-Size Distributions | | | SOM |
| | | | | | Sand | Silt | Clay | |
|---|---|---|---|---|---|---|---|---|
| *U. pumila* | 38 | 16.60 | 120.63 | 0.31 | 39.46 | 50.85 | 9.69 | 1.34 |
| *C. korshinskii* | 20 | 2.38 | 2.65 | 0.43 | 52.89 | 39.25 | 7.86 | 0.85 |

Note: DBH = diameter at breast height; SBD = stem basal diameter; SA = sapwood area; LAI = leaf area index; SOM = soil organic matter.

## 2.2. Measurement of Sap Flux Density and Estimation of Canopy Transpiration

Measurements of sap flow were carried out using Granier-type thermal dissipation probes (TDPs) between 1 May and 30 September in 2020 and 2021 [30]. The sensors consisted of a pair of probes having a 20 mm length and 2 mm diameter, each of which had a cooper constant thermocouple. Based on the distributions of DBH and SBD, six sampled trees in the *U. pumila* stand and six sampled stems in the *C. korshinskii* stand were selected for sap flux measurements. The probes were inserted into the north-facing side of the truck (stem) at a height of 1.3 m (sampled *U. pumila*) and 15 cm (sampled *C. korshinskii*) above ground level to minimize exposure to the sun. In order to protect the probes against physical damage, solar radiation, temperature fluctuations, and rain, they were mounted in waterproof silicone and covered with an aluminum cover. The temperature differences between heated and reference probes were recorded on a data logger at 10 min intervals

(CR1000, Campbell Scientific Inc., Logan, UT, USA). The sap flux density ($F_d$) was estimated using the standard calibration for the TDP method:

$$F_d = 0.714 \times \left( \frac{\Delta T_{max} - \Delta T}{\Delta T} \right)^{1.231} \tag{1}$$

where $F_d$ (mL cm$^{-2}$ min$^{-1}$) is the sap flux density, $\Delta T$ (°C) is the temperature difference between the two probes at any given time, and $\Delta T max$ (°C) is the maximum temperature difference between sensors when the sap flux density is close to zero. Due to the fact that we only used one pair of probes per sample tree, azimuthal and radial variations in sap flux within trees were ignored In calculating canopy transpiration per area of ground ($E_c$, mm h$^{-1}$); we multiplied the mean $F_d$ for each sample tree ($F_d$-avg) by the total sapwood area of the study plot per unit area of ground:

$$E_c = F_{d-avg} \times \frac{A_{si}}{A_g} \times 600 \tag{2}$$

where $F_d$-avg is the average sap flux density of sampled trees (or shrubs), $A_g$ is the ground surface area of the studied plots (400 m$^2$), and $A_{si}$ is the total sapwood area in the studied plot. In order to measure the sapwood area of *U. pumila*, the staining method was applied to 19 trees both inside and outside the plot. Exponential growth function relationships between sapwood and DBH ($A_s = 27.10 \times e^{0.09DBH}, n = 19, R^2 = 0.83$) were established. Furthermore, we used the stem basal cross-sectional area as the scalar to estimate the $E_c$ and stand transpiration for the *C. korshinskii* stand. The total sapwood area in the plot was 6016.04 cm$^2$ and 6877.70 cm$^2$ for the *U. pumila* and *C. korshinskii* stands, respectively.

### 2.3. Meteorological Data and Soil Water Measurement

Precipitation ($P$, mm), air temperature ($T_a$, °C), relative humidity ($RH$, %), wind speed ($u_2$, m s$^{-1}$), and photosynthetically active radiation ($PAR$, μmol m$^{-2}$ s$^{-1}$) were measured continuously with an Onset HOBO U21 automatic weather station (Onset Computer Corp., Bourne, MA, USA) located in the open areas neighboring the *U. pumila* plantation stand. Air temperature and relative humidity were measured at a height of 1.5 m, and precipitation, wind speed, and photosynthetically active radiation were measured at a height of 2.0 m (Figure 1). These variables were recorded every 10 min as averages using data loggers of the CR1000 series (Campbell Scientific, Logan, UT, USA). *VPD* (kPa) was derived from air temperature and the relative humidity:

$$VPD = 0.611 \times \exp\left( \frac{17.502 T_a}{T_a + 240.97} \right) \times (1 - RH) \tag{3}$$

The soil volumetric water content ($\theta$, %) was continuously monitored at a single location in *U. pumila* and *C. korshinskii* stands with an EC–5TE sensor (Decagon, Inc. Pullman, WA, USA). Both EC–5TE sensors were installed at depths of 0–10 cm, 10–20 cm, 20–40 cm, 40–60 cm, and 60–100 cm below the ground surface. Measurements were taken every 10 s, and the 30 min averages were recorded by CR1000 data loggers (Campbell Scientific, Logan, UT, USA). The soil water content values used in the paper were averaged for the depths of 0–100 cm for each plot. Relative extractable soil water (REW) was calculated using averaged $\theta$ across 0–100 cm:

$$REW = \frac{\theta - \theta_{min}}{\theta_{max} - \theta_{min}} \tag{4}$$

where $\theta_{min}$ and $\theta_{max}$ are the minimum and maximum daily average soil water content, respectively, during May–September in 2020 and 2021.

To account for climate impacts on stand canopy transpiration, daily potential evapotranspiration ($PET$, mm d$^{-1}$) was calculated by the Penman–Monteith equation based on FAO–56 [31]:

$$PET = \frac{0.408\Delta(R_n - G) + \gamma\left(900/(T_a + 273)\right)u_2 VPD}{\Delta + \gamma(1 + 0.34u_2)} \tag{5}$$

where $R_n$ is the net radiation (MJ m$^{-2}$ d$^{-1}$), $G$ is the soil heat flux density ($G \approx 0$ for daily scale), $\gamma$ is the psychrometric constant (kPa °C$^{-1}$), and $\Delta$ is the slope of the vapor pressure curve (kPa °C$^{-1}$).

### 2.4. Canopy Conductance and Decoupling Coefficient

$G_c$ (mm s$^{-1}$) was calculated by multiplying the conductance coefficient ($K_G$) by $E_c$ and dividing the result by $VPD$ [32]:

$$G_c = \frac{K_G \times E_c}{VPD} \tag{6}$$

where $K_G$ is the conductance coefficient as a function of air temperature ($115.8 + 0.4236\, T_a$, kPa m$^3$ kg$^{-1}$), which accounts for the temperature effect on the psychometric constant, the latent heat of vaporization, and the specific heat and density of air.

We estimated the decoupling coefficient ($0 < \Omega < 1$) as a means of quantifying the degree of coupling between the canopy and atmosphere. Canopies are aerodynamically well coupled to the atmosphere when $\Omega$ approaches 0, and canopy transpiration is determined primarily by stomatal opening. As $\Omega$ approaches 1, stomatal control on canopy transpiration becomes weaker, and canopy transpiration is more influenced by solar radiation. The $\Omega$ was estimated as follow [33]:

$$\Omega = \frac{1}{1 + [\gamma/(\Delta + \gamma)]\left(g_a/G_c\right)} \tag{7}$$

where $g_a$ is the aerodynamic conductance (m s$^{-1}$), calculated as follows:

$$g_a = \frac{1 + 0.54 U_z}{\left[\ln\,(z - d)/z_0\right]^2} \tag{8}$$

where $U_z$ is the wind speed above the canopy (m s$^{-1}$), which is derived from the measured wind speed at a 2.0 m height ($u_2$). $z$ is usually equal to the canopy height (m), $z_0$ is the roughness height (usually 0.1 * H, where H is the canopy height) and $d$ is the displacement height (0.75 * H).

### 2.5. Data Analysis

In the present study, significant differences in daily $E_c$, $G_c$, and $\Omega$ between the two growing seasons and the two planted forests were tested using two-way repeated measures ANOVA. The differences in the micrometeorological variables during the two growing seasons of 2020 and 2021 were testing using a paired samples t-test. Then, the relationships between $PAR$, $VPD$, and $E_c$ were mainly analyzed using an exponential threshold function after partitioning the data into two classes (stressed water conditions: REW < 0.40 and non-stressed water conditions: REW > 0.40), as follows:

$$E_c = a\left(1 - e^{-bx}\right) \tag{9}$$

where $a$ and $b$ are the fitting parameters, and $x$ is the corresponding meteorological variable.

The response of $G_c$ to $VPD$ was quantified using a linear logarithmic function [34]:

$$G_c = -m \ln VPD + G_{ref} \tag{10}$$

where $m$ is the canopy conductance's sensitivity to $VPD$ and $G_{ref}$ is the reference canopy conductance when $VPD = 1$ kPa. An upper boundary line was derived based on the data of

$E_c$ at least one standard deviation greater than the mean $E_c$ of each *VPD* (0.5 kPa) interval, from which the parameters of m and $G_{ref}$ were determined.

All statistical analyses were performed using the SPSS 21 software program (SPSS Inc., Chicago, IL, USA) and all the figures were created using Sigmaplot 11.0 software (Hearne Scientific Software Plc, Melbourne, Australia).

## 3. Results

### 3.1. Environmental Variables

During the growing season (May–September), the total precipitation was 256 and 283 mm in 2020 and 2021, respectively (Figure 2), accounting for 88.90% and 98.31% of the long-term average precipitation (288 mm, 1980–2020). The photosynthetically active radiation was $402.47 \pm 124.65$ µmol m$^{-2}$ s$^{-1}$ and $442.56 \pm 126.41$ µmol m$^{-2}$ s$^{-1}$, whereas *PET* averaged $3.65 \pm 1.36$ mm d$^{-1}$ and $3.72 \pm 1.30$ mm d$^{-1}$ during the growing season in 2020 and 2021, respectively (Figure 2a). During both growing seasons, daily air temperatures displayed marked seasonal variations, with a range of 5.1 to 22.6 °C and 1.5 to 23.1 °C, and mean values of $15.14 \pm 3.88$ °C and $15.02 \pm 4.28$ °C, in 2020 and 2021, respectively (Figure 2b). The average *VPD* was $0.63 \pm 0.35$ kPa and $0.61 \pm 0.32$ kPa, whereas $u_2$ was $3.11 \pm 1.56$ m s$^{-1}$ and $3.31 \pm 1.77$ m s$^{-1}$ in 2020 and 2021, respectively (Figure 2c). The mean soil water content at the 0–100 cm soil layer was 8.03% for *U. pumila* and 5.09% for *C. korshinskii* in 2020, and 10.42% for *U. pumila* and 5.33% for *C. korshinskii* in 2021. The soil water content in the 0–40 cm layer increased quickly after precipitation events, whereas that of the 40–100 cm layer was stable during the growing season. In the vertical profile, the soil water content in the 40–100 cm layer was lower than that in other profiles in *U. pumila*, whereas the lowest soil water content occurred at 20–40 cm in *C. korshinskii* (Figure 3).

### 3.2. Canopy Transpiration per Unit Ground Area

During the measurement period, $E_c$ for the *U. pumila* stand ranged from 0.08 to 1.77 mm d$^{-1}$, with a mean value of $0.55 \pm 0.34$ mm d$^{-1}$ in 2020; whereas, in 2021, $E_c$ ranged from 0.08 to 2.57 mm d$^{-1}$, with a mean value of $0.66 \pm 0.32$ mm d$^{-1}$. For *C. korshinskii*, $E_c$ ranged from 0.10 to 1.41 mm d$^{-1}$ and from 0.21 to 1.30 mm d$^{-1}$, with mean values of $0.74 \pm 0.26$ mm d$^{-1}$ and $0.77 \pm 0.24$ mm d$^{-1}$ in 2020 and 2021, respectively (Figure 4a,c). The accumulated $E_c$ over the entire growing season was 83.72 and 113.65 mm for *U. pumila* in 2020 and 2021, respectively, accounting for 32.70% and 40.14% of the precipitation over the same period. The accumulated $E_c$ was 101.29 and 117.77 mm for the *C. korshinskii* stand in 2020 and 2021, respectively, accounting for 39.57% and 41.60% of the precipitation over the same period (Figure 4b,d).

### 3.3. Response of Canopy Transpiration to Environmental Variables

Statistical analysis indicated that the positive effect of environmental variables on $E_c$ had a ranking of $\theta > T_a > PAR > u_2 > VPD > PET$ in 2020 and $PAR > u_2 > \theta > T_a > P > PET$ in 2021 for *U. pumila*. The positive effect of environmental variables on $E_c$ had a ranking of $PAR > \theta > u_2 > T_a > VPD > P$ in 2020 and $P > \theta > T_a > VPD > PAR > u_2 > PET$ in 2021 for *C. korshinskii* (Table 2). This suggests that *PAR*, $\theta$, and *VPD* were major determinants of transpiration for both *U. pumila* and *C. korshinskii*.

The $E_c$ responded to REW following a quadratic polynomial function for *U. pumila* both in 2020 and 2021 (Figure 5a,b). $E_c$ reached the maximum values when REW = 0.4–0.5 in 2020 and REW = 0.4 in 2021. The $E_c$ responded to REW following a saturated exponential function for *C. korshinskii* both in 2020 and 2021 (Figure 5c,d). $E_c$ rapidly increased with rising REW when REW < 0.4, $E_c$ then increased slowly when REW > 0.4, and, finally, $E_c$ tended to be saturated at 1.31 and 1.14 mm when REW was close to 1 in 2020 and 2021, respectively. Additionally, the response of $E_c$ to PAR and VPD was affected by different soil water conditions. The $E_c$ increased with an increasing *PAR* and *VPD* for *U. pumila* under REW < 0.40, while it tended to be saturated when *VPD* reached 2.0 kPa (Figure 6a,b). *PAR* and *VPD* explained 28% and 22% of the variation in $E_c$ for *U. pumila* under REW < 0.40,

respectively, whereas $E_c$ maintained a high value and stability when REW > 0.4. The $E_c$ increased significantly with an increasing *PAR* and *VPD* for *C. korshinskii* under different soil water conditions (Figure 6c,d). The $E_c$ exhibited an exponentially saturating response to *VPD* and tended to level off at 1.5 kPa under REW < 0.4, and there was no obvious threshold value of $E_c$ under REW > 0.4. *VPD* explained 16% and 81% of the variation in $E_c$ for *C. korshinskii* under REW < 0.40 and REW > 0.4, respectively, whereas *PAR* explained 23% and 52% of the variation in $E_c$ for *C. korshinskii* under REW < 0.40 and REW > 0.4, respectively.

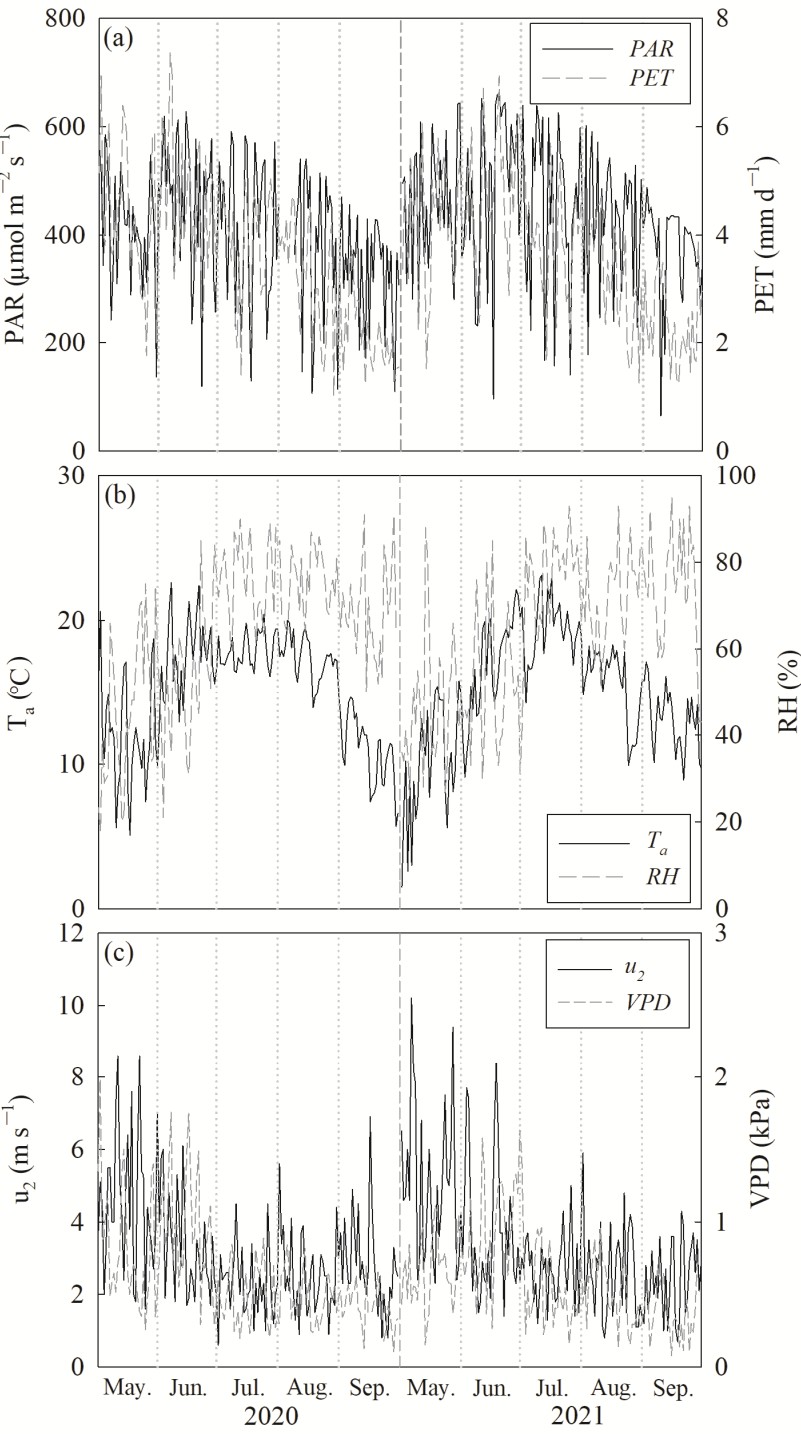

**Figure 2.** Daily variations in (**a**) photosynthetically active radiation (*PAR*) and daily potential evapotranspiration (*PET*), (**b**) temperature ($T_a$) and relative humidity (*RH*), (**c**) wind speed ($u_2$) and vapor pressure deficit (*VPD*).

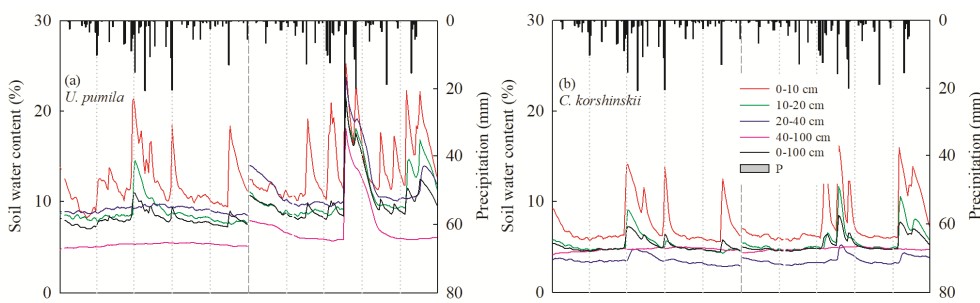

**Figure 3.** Daily precipitation and volumetric soil water content (0–100 cm) from May 1 to September 30 in (**a**) *U. pumila* and (**b**) *C. korshinskii*.

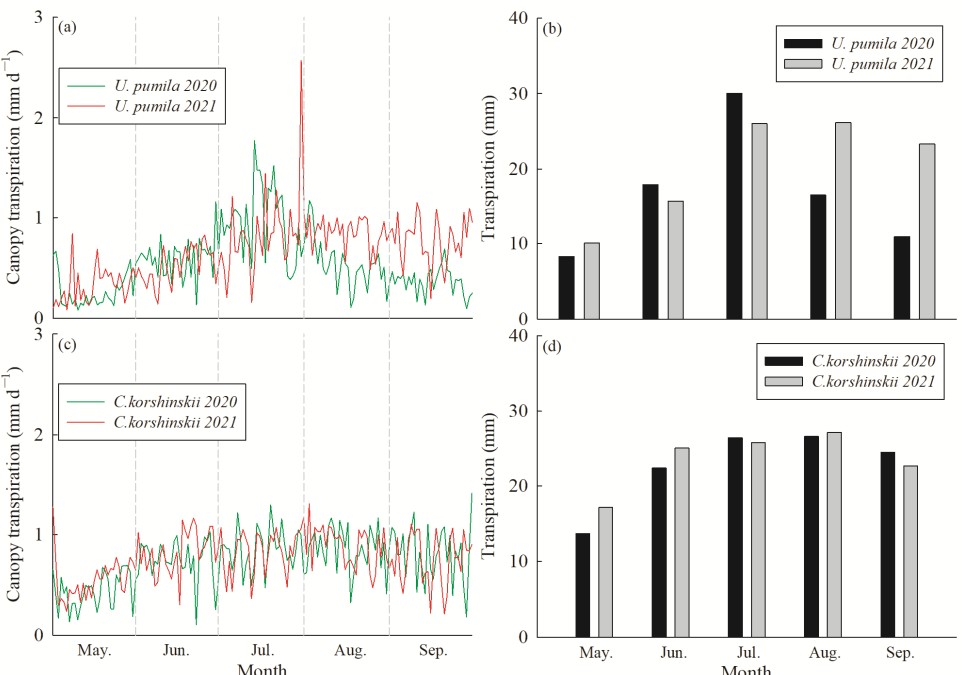

**Figure 4.** Variations in daily canopy transpiration of per ground area ($E_c$, **a,c**), and monthly transpiration (**b,d**) for *U. pumila* and *C. korshinskii*.

**Table 2.** Correlation coefficients between canopy transpiration ($E_c$) and environmental factors for *U. pumila* and *C. korshinskii* in 2020 and 2021.

|  |  | *PAR* | $T_a$ | *RH* | *VPD* | $u_2$ | *p* | *PET* | $\theta$ |
|---|---|---|---|---|---|---|---|---|---|
| *U. pumila* | *Ec* (2020) | 0.522 ** | 0.615 ** | - | 0.366 ** | −0.404 * | - | 0.277 ** | 0.691 ** |
|  | *Ec* (2021) | 0.513 ** | 0.369 ** | - | - | −0.508 ** | −0.253 ** | 0.176 * | 0.502 ** |
| *C. korshinskii* | *Ec* (2020) | 0.475 ** | 0.362 ** | - | 0.354 ** | −0.462 ** | −0.262 ** | - | 0.474 ** |
|  | *Ec* (2021) | 0.349 ** | 0.402 ** | - | 0.389 ** | −0.330 ** | −0.497 ** | 0.185 * | 0.444 ** |

Note: $PAR$ = photosynthetically active radiation; $T_a$ = air temperature; $RH$ = relative humidity; $VPD$ = vapor pressure deficit; $u_2$ = wind speed; $P$ = precipitation; $PET$ = daily potential evapotranspiration; $\theta$ = soil water content. ** indicates $p < 0.01$, * indicates $p < 0.05$.

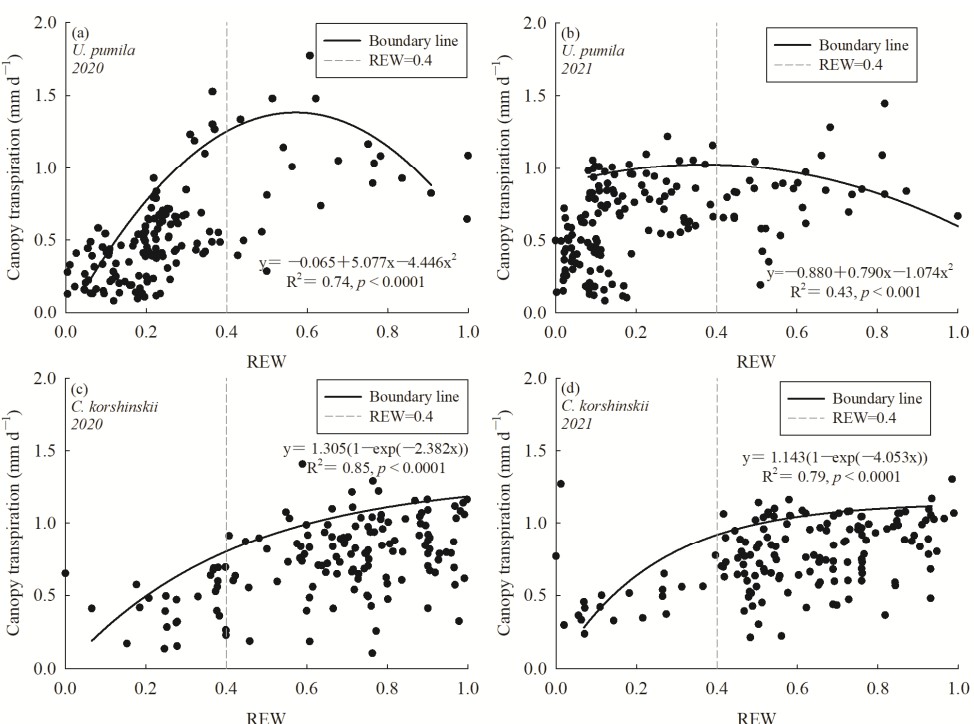

**Figure 5.** Response of canopy transpiration per ground area to the relative extractable soil water and boundary line analysis for *U. pumila* and *C. korshinskii* in 2020 (**a**,**c**) and 2021 (**b**,**d**).

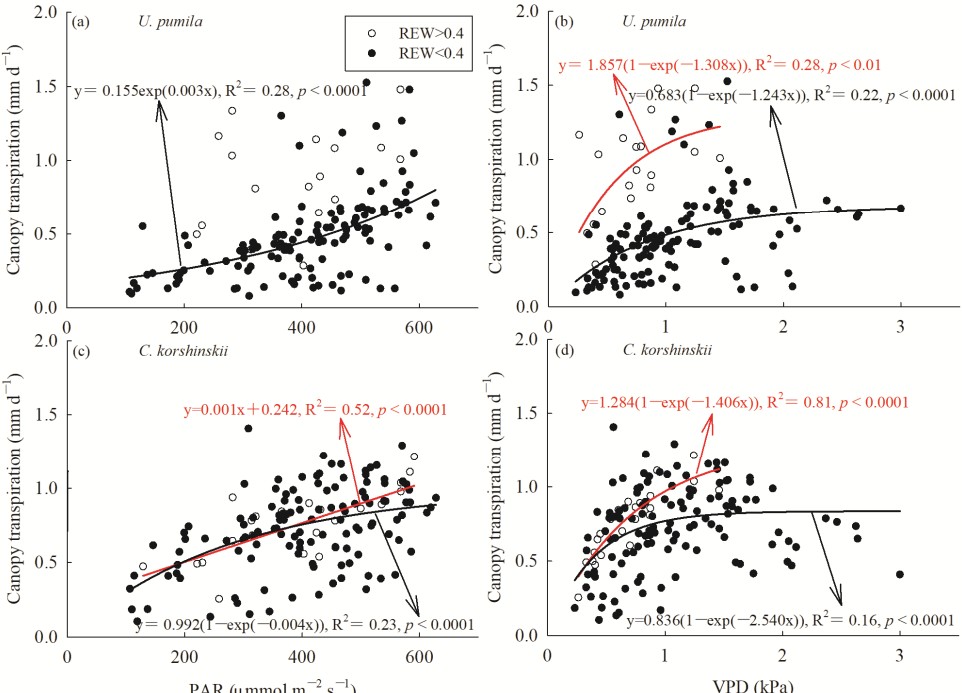

**Figure 6.** Relationships between canopy transpiration per ground area and daily environmental variables for *U. pumila* and *C. korshinskii* under stressed (REW < 0.40) and non-stressed (REW > 0.4) conditions across the measurement periods of 2020: (**a**,**c**) photosynthetically active radiation; (**b**,**d**) vapor pressure deficit.

### 3.4. Response of Canopy Stomatal Conductance to VPD

The $G_c$ showed similar seasonal variations for *U. pumila* and *C. korshinskii* during 2020 and 2021. The maximum monthly $G_c$ of *U. pumila* was $1.98 \pm 1.07$ mm s$^{-1}$ and

$2.02 \pm 1.42$ mm s$^{-1}$ in July during 2020 and 2021, respectively. However, in both years, the $G_c$ of *C. korshinskii* was significantly higher in July–September than in May–June ($F = 15.517$, $p < 0.001$ in 2020 and $F = 11.477$, $p < 0.001$ in 2021) (Figure 7). The $G_c$ significantly decreased with an increase in *VPD* for both afforestation species (Figure 8). *VPD* explained 15% and 41% of total variation in $G_c$ for *U. pumila* in 2020 and 2021, respectively (Figure 8a,b), and 45% and 51% of total variation in $G_c$ for *C. korshinskii* in 2020 and 2021, respectively (Figure 8c,d). In addition, *m* had a significantly higher effect in 2021 than in 2020 for *U. pumila* (3.01 > 2.50) and *C. korshinskii* (2.83 > 1.96) after applying boundary line analysis, whereas no significant difference in the reference canopy stomatal conductance ($G_{ref}$) was observed for the two planted species. Moreover, the higher $G_c$ of *U. pumila* and *C. korshinskii* when REW > 0.4 than when REW < 0.4 (*U. pumila*: 2.46 > 0.78, $F = 26.383$, $p < 0.001$ in 2020; *C. korshinskii*: 2.02 > 1.36, $F = 14.359$, $p < 0.001$ in 2021) indicates the significant effect of soil water content on canopy stomatal conductance. *m* was also higher under REW > 0.4 conditions than under REW < 0.4 conditions for both of the planted species (*U. pumila*: 3.30 > 0.87 in 2020 and 3.59 > 2.98 in 2021; *C. korshinskii*: 2.26 > 1.97 in 2020 and 3.70 > 2.08 in 2021) (Figure 8e–h).

Throughout the measurement period, the daily $\Omega$ of *U. pumila* ranged from 0.0001 to 0.0112, with mean values of $0.0015 \pm 0.0014$ and $0.0020 \pm 0.0019$ in 2020 and 2021, respectively (Figure 7). The daily $\Omega$ of *C. korshinskii* ranged from 0.0003 to 0.0145, with mean values of $0.0024 \pm 0.0016$ and $0.0024 \pm 0.0018$ in 2020 and 2021, respectively. No differences were found for the value of $\Omega$ between 2020 and 2021 for both *U. pumila* ($F = 5.896$, $p = 0.016$) and *C. korshinskii* ($F = 0.021$, $p = 0.884$).

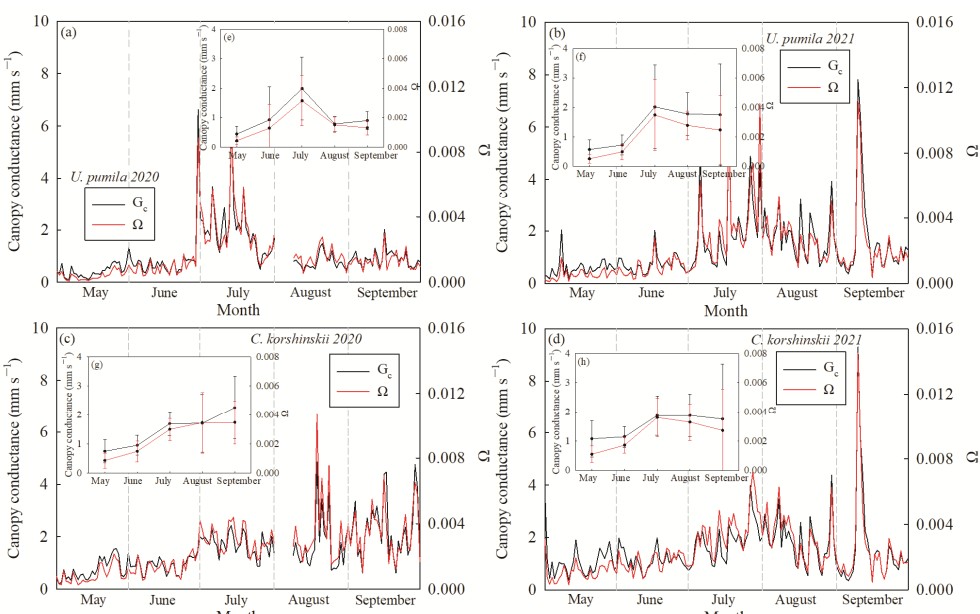

**Figure 7.** Seasonal variations in daily mean canopy conductance per ground area ($G_c$) and mean decoupling coefficient ($\Omega$) for *U. pumila* and *C. korshinskii* during the growing season of 2020 (**a,c**) and 2021 (**b,d**); monthly $G_c$ and $\Omega$ for *U. pumila* and *C. korshinskii* of 2020 (**e,g**) and 2021 (**f,h**).

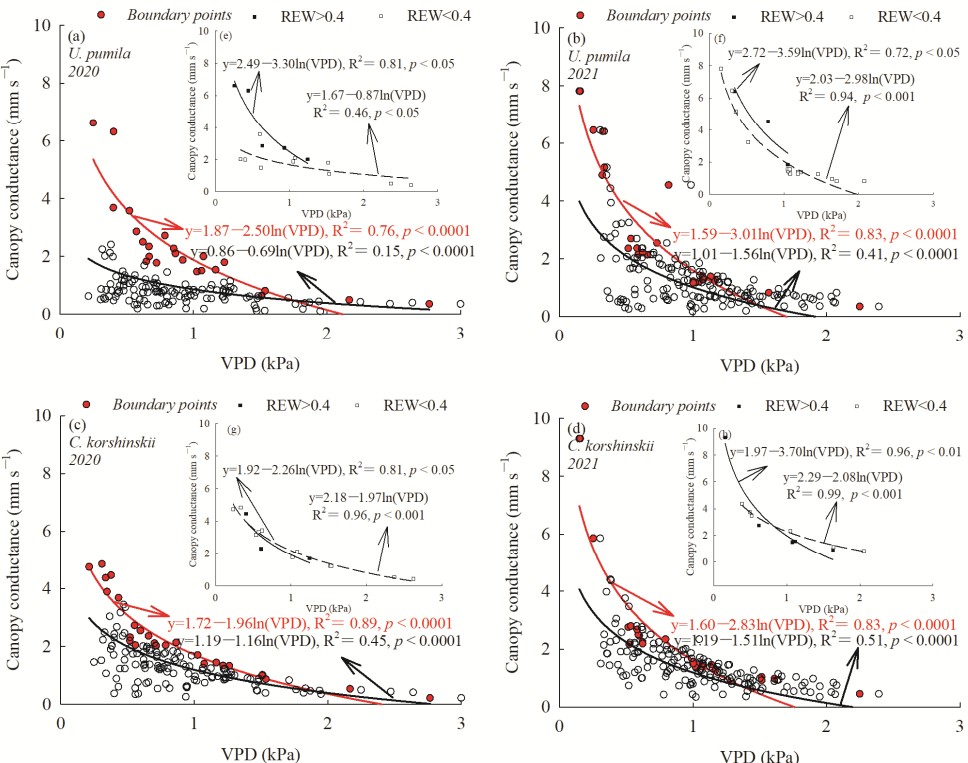

**Figure 8.** Relationships between canopy conductance per ground area and daily vapor pressure deficit in the measurement periods (black line) and boundary line analysis (red line) for *U. pumila* and *C. korshinskii* in 2020 (**a**,**c**) and 2021 (**b**,**d**); boundary line analysis under stressed (REW < 0.4) and non-stressed (REW > 0.4) conditions for *U. pumila* and *C. korshinskii* in 2020 (**e**,**g**) and 2021 (**f**,**h**).

## 4. Discussion

### 4.1. Canopy Transpiration of U. pumila and C. korshinskii

The $E_c$ of the *U. pumila* stand in this study was 0.55 and 0.66 mm d$^{-1}$, which is similar to the values reported for Mongolian pine planted in the southeastern Keerqin Sandy Land [35] and half-mature Mongolian pine planted in the Rare Psammophytes Protection Botanical Base [36], but significantly lower than those of Chinese pine (0.7 mm d$^{-1}$), Mongolian pine (1.1 mm d$^{-1}$), and *Populus × xiaozhuanica* plantations (1.2–1.5 mm d$^{-1}$) planted in the Keerqin Sandy Land [16]; *Populus simonii* (175.2 kg d$^{-1}$) planted on the Loess Plateau [37]; and *Populus euphratica* Oliv. planted in the lower reaches of the Heihe River Basin [38]. The $E_c$ of the *C. korshinskii* stand in this study was 0.74 and 0.77 mm d$^{-1}$ in the two seasons, respectively. These values are lower than those for the same species planted in the Loess Plateau under different precipitation regions [37,39]. Compared with other major planted shrubs, a similar range of transpiration was reported for a *Vitex negundo* L. plantation (107.21 mm cumulative in the growing season), but a higher transpiration of *Hippophae rhamnoides* L. was found in the Loess Plateau (122.33–180.28 mm cumulative in the growing season) [11]. This may be related to the higher evaporative demand in drier and colder regions and the lower soil water availability in the Bashang Plateau. The mean annual precipitation in our study site was 100–150 mm lower than that for planted forests in the Keerqin Sandy Land and the Loess Plateau. In addition, a relatively lower soil water content may also increase the hydraulic resistance of the soil–root system, preventing water movement between soil and plant leaves, which ultimately decreases plantations' transpiration rate [40]. Finally, short heights and small sapwood areas, resulting from severe canopy dieback and planted forest degradation, were found for the *U. pumila* stand in this study (Table 2), and canopy transpiration was thus exhibited at a lower level.

$E_c$ responded to REW as a saturated exponential function for *C. korshinskii* in this study (Figure 5c,d), which was in agreement with results reported for broadleaved and coniferous

trees under different climates [41], a larch plantation in the semiarid Northwest China [42], a black locust plantation [43], and *C. korshinskii* [44] on the Loess Plateau. However, the REW threshold in this study is similar to or relatively higher than those in the studies reported above. Li et al. indicated that the REW threshold is generally higher under dry conditions than under wet conditions [23]. Averaged soil water content of the *C. korshinskii* stand was 5.09% in 2020 and 5.33% in 2021, which was lower than those in the plantations reported above (Figure 3b). $E_c$ presented a polynomial pattern in REW for *U. pumila* in this study (Figure 5a,b), which was consistent with that in *Haloxylon ammodendron* and *Calligonum mongolicum* in Northwest China [45]. The reason that $E_c$ was lower during the higher REW period is linked to the much lower *VPD* and $T_a$ when compared to the dry period (Figures 2 and 6b). The daily $E_c$ of both the *U. pumila* and *C. korshinskii* stands increased with increasing *PAR* and *VPD*, and $E_c$ tended to be saturated at high *VPD* values. These results are similar to those reported for poplar trees [16], Mongolian pine [46], and other planted species [12]. In both plantations, however, the *VPD* threshold decreased as soil water content decreased, ranging from non-stressed to water stressed. An obvious *VPD* threshold was observed at approximately 2.0 kPa for the *U. pumila* stand and 1.5 kPa for the *C. korshinskii* stand when REW < 0.4. Liu et al. indicated that the utilization of surface soil water gradually increased with the increase in the degree of degradation of the agroforestry shelterbelts [47]. We then concluded that the *U. pumila* and *C. korshinskii* stands primarily obtained water from the shallow and middle soil layers because of the root distribution pattern and the degree of canopy dieback. As soil water decreased, the increasing whole-tree hydraulic resistance triggered canopy stomatal closure, and hence limited the response of transpiration to major environmental variables. In addition, the $\Omega$ values (Figure 6) were also lower than those reported for mature Mongolian pine (0.041), half-mature Mongolian pine (0.15), and young Mongolian pine (0.18) [36], and *Schima superba* (0.22) [48]. The smaller LAI of the *U. pumila* (0.31) and *C. korshinskii* (0.43) plantations in this study, compared to that of the Mongolian pine (0.96–1.54) in the Horqin desert and the Mu Us desert, indicates that decreasing the canopy resistance resulted in strong canopy–atmosphere coupling. Therefore, under high *VPD* conditions, canopy stomata must be closed to prevent water potential from dropping below the threshold and thus inhibiting tree transpiration. Moreover, a relatively lower *VPD* threshold under water stress was observed for both *C. korshinskii* (1.5 kPa) and *U. pumila* (2.0 kPa), indicating that the transpiration process of the two species was affected significantly by soil drought [49]. Planted species usually have a shallower fine root distribution and lower stem and leaf hydraulic conductivity [38,50]. As concluded above, the transpiration process and water-use strategy of *U. pumila* and *C. korshinskii* were revealed in the Bashang Plateau, and further research on the water-use pattern of typical planted forests should be undertaken in the future. However, in the present study, we did not investigate seasonal variations in LAI and its effects on canopy transpiration during the growing season for *U. pumila* and *C. korshinskii*. Additional studies on canopy transpiration and stomatal conductance combined with different plantation ages and densities would provide further insights into the ecohydrological processes and afforestation management on the Bashang Plateau.

*4.2. Canopy Stomatal Conductance of U. pumila and C. korshinskii*

The $G_c$ values were significantly higher under non-stressed conditions than under stressed conditions for both plantations, indicating the strong limitations of soil water on canopy stomatal conductance. In addition, the significantly nonlinear negative relationship between $G_c$ and *VPD* in both the *U. pumila* and *C. korshinskii* plantations suggested that stronger stomatal regulation of canopy transpiration responded to higher *VPD*. Stomatal closure thus played an important role in maintaining water status to avoid a catastrophic loss of xylem function [51]. This drought adaptation mechanism can be characterized as a water-saving strategy, consistent with those reported for *P. sylvestris* plantations [36,52,53].

We also found that the parameters of *m* and $G_{ref}$ for the *U. pumila* plantation were higher than those for the *C. korshinskii* plantation (Figure 8). This was contrary to the

findings of Song et al. [46], who observed that, in comparison with Mongolian pine (an introduced species), Chinese pine (a native species) had relatively higher values of $m$ and $G_{ref}$. We further suggest that $m$ and $G_{ref}$ for both the *U. pumila* and *C. korshinskii* plantations were higher under REW > 0.4 than under REW < 0.4 (Figure 8e–h), indicating that the sensitivity of canopy stomatal conductance to *VPD* decreased with soil drought; this was consistent with the findings of Novick et al. [54], Jiao et al. [19] on the Loess Plateau, and Song et al. [35] in the Horqin desert. Soil drought can negatively affect the hydraulic conductance of soil-to-plant leaf transporting paths and decrease $m$ values, thus limiting the transpiration rate. Grossiord et al. indicated that reduced soil water availability negated the benefits of stomatal and hydraulic adjustments and resulted in reduced transpiration in juniper [55]. Similar results were found for grapevines, both in pot grown and field-grown experiments [56]. Therefore, further decreasing hydraulic conductance to prevent xylem cavitation as soil water stress increases results in a less sensitive response of $G_c$ to *VPD*. This may be one of the important reasons why planted forests are vulnerable to dieback and degradation. Further research on the hydraulic traits of typical planted forests should be undertaken in the future.

## 5. Conclusions

In the present study, canopy transpiration and stomatal conductance were estimated for an *U. pumila* and a *C. korshinskii* stand. During the growing season, canopy transpiration was 83.72 and 113.65 mm in 2020 and 2021, respectively, for the *U. pumila* stand, and 101.29 and 117.77 mm in 2020 and 2021, respectively, for the *C. korshinskii* stand. Transpiration of *U. pumila* and *C. korshinskii* were better correlated to soil water content, photosynthetically active radiation, and *VPD*. Both planted species showed reduced *VPD* sensitivity of canopy transpiration as soil water decreased, indicating the transpiration process was affected significantly by soil drought. The *VPD* threshold was 1.50 and 2.0 kPa for *U. pumila* and *C. korshinskii*, respectively. Furthermore, the transpiration of both planted species was mainly regulated by stomatal opening due to low values of the decoupling coefficients (0.0015 and 0.0020 for *U. pumila*; 0.0024 for *C. korshinskii*). Both *U. pumila* and *C. korshinskii* gradually displayed a reduced canopy stomatal conductance with increasing *VPD*, but decreasing stomatal sensitivity to *VPD* in response to soil drought was also observed. Based on the results above, we conclude that both *U. pumila* and *C. korshinskii* plantations exhibited water-saving strategies under a cold and arid environment on the Bashang Plateau. These findings provide a deeper understanding of transpiration dynamics of different forest functional types on the Bashang Plateau, which may be applied for afforestation management in semiarid regions.

**Author Contributions:** Conceptualization, Y.Z.; methodology, Y.Z. and W.L.; investigation, J.Z. and N.W.; data curation, H.Y. and B.X.; Supervision, X.W.; Writing—original draft, Y.Z.; Writing—review and editing, Y.Z. and W.L. All authors have read and agreed to the published version of the manuscript.

**Funding:** This research was funded by the National Natural Science Foundation of China (No. 42101019, 42001027, 51909052), Natural Science Foundation of Hebei Province (D2019205274, D2019403115, D2021403023), Science and Technology Project of Hebei Education Department (QN2019152, BJK2022022), Key Laboratory of Agricultural Water Resources & Hebei Key Laboratory of Agricultural Water-Saving, Center for Agricultural Resources Research, Institute of Genetics and Developmental Biology (KFKT201903), Science Foundation of Hebei Normal University (L2018B22), Special soft science research project of Hebei Provincial Science and Technology Plan (21557401D).

**Informed Consent Statement:** Not applicable.

**Acknowledgments:** We thank the reviewers and editors for their work.

**Conflicts of Interest:** The authors declare no conflict of interest.

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
