# Peer review of "Canopy Transpiration and Stomatal Conductance Dynamics of Ulmus pumila L. and Caragana korshinskii Kom. Plantations on the Bashang Plateau, China"

_forests, doi:10.3390/f13071081_

Round 1

Reviewer 1 Report

Two different forest types in the Bashang Plateau, China, are examined regarding the transpiration dynamics and canopy transpiration (Ec) correlation to key environmental variables (PAR, VPD, ETo). The manuscript (ms) is well written, patterns were detected which will provide useful contribution to the field of planted forest management. Moreover, the study adds information to the theoritical knowledge of plantation responses to water stress under specific conditions. 

The english is fluent, however, some typos and few syntax errors need editing (see the comments in the attached pdf file).

My suggestions for the ms:

Lines 56-59: Citation(s) needed here to support the assertion (see the attached pdf file).

Lines 62-64: the definition needs to be revised and more specific to avoid being confused with the more general concept of evapotranspiration.

Line 96: Gc: Please define the symbol the first time it is mentioned in the main text (see also line 213 in the pdf fi).

Lines 125-126 and Figure 1.: Please edit Figure 1 by embedding an additional figure-map of the whole China (or a large part of it) where the study area is clearly marked (e.g. pinned), add some indicative city names around as well, because it s hard for international readers to follow. Then adjust the caption of Figure 1 accordingly.

Table 1: Caption: Please prefer the term "detailed information" instead.

"SOM" also needs explanation in Table's Note.

Lines 186, 265, 275 and 228-231: wind speed is somewhat confusing in the ms. The wind speed needs some clarification regarding the height of its measurement throughout the ms. Consider to use differentiated abbreviations when the height of measurement differs (e.g. u2 for ws measured at 2 m height and so on), or define if canopy wind speed uz is the same as u throughout the ms.

Lines 249-250: Elaborate a little on ANCOVA to justify its usage. Mention which one is the dependent variable and which one is the factor etc, in few lines.

Lines 259-260: Is ETo in mm/day? Please edit the units to also depict the time scale.

Table 2: Why precipitation is not included to the environmental factors for which correlation coefficients are sought? Consider to include it.

Moreover, the explanation of θ symbol is missing from the Note. Please add it.

At last, what are the limitations of your experiment (design, methods, materials etc)/study that could be also referred/discussed? Add few lines in the Discussion.

Kind regards

Reviewer 2 Report

Dear Authors,

After carefully reading your manuscript, I suggest the following improvements;

1. Figure1, include a national map clearly locating the North China agro-pastoral ecotone.

2. Please check all the equations in the manuscript to verify if all the variables are correctly described, as well as their units in a compatible way.

3. Lines 259-261. Reference evapotranspiration should be in units of mm/day.

4. Line 265 and Figure 2.  wind speed units should be m/s not mm/s.

5. Figure 3. Complete the figure with the precipitation units on the right vertical axis.
